# The Role of Dietary Protein in Body Weight Regulation among Active-Duty Military Personnel during Energy Deficit: A Systematic Review

**DOI:** 10.3390/nu15183948

**Published:** 2023-09-12

**Authors:** Robert E. Anderson, Shanon L. Casperson, Hannah Kho, Kyle D. Flack

**Affiliations:** 1Department of Nutrition, Gillings School of Global Public Health, University of North Carolina at Chapel Hill, Chapel Hill, NC 27517, USA; 2Grand Forks Human Nutrition Research Center, Agricultural Research Service, U.S. Department of Agriculture, Grand Forks, ND 58203, USA; 3Department of Dietetics and Human Nutrition, College of Agriculture, Food and Environment, University of Kentucky, Lexington, KY 40506, USA

**Keywords:** active-duty military, protein, body weight, energy deficit, fat mass, fat-free mass, body composition

## Abstract

Active-duty military personnel are subjected to sustained periods of energy deficit during combat and training, leaving them susceptible to detrimental reductions in body weight. The importance of adequate dietary protein intake during periods of intense physical training is well established, where previous research has primarily focused on muscle protein synthesis, muscle recovery, and physical performance. Research on how protein intake may influence body weight regulation in this population is lacking; therefore, the objective of this review was to evaluate the role of dietary protein in body weight regulation among active-duty military during an energy deficit. A literature search based on fixed inclusion and exclusion criteria was performed. English language peer-reviewed journal articles from inception to 3 June 2023 were selected for extraction and quality assessment. Eight studies were identified with outcomes described narratively. The study duration ranged from eight days to six months. Protein was directly provided to participants in all studies except for one. Three studies supplied additional protein via supplementation. The Downs and Black Checklist was used to assess study quality. Five studies were classified as good, two as fair, and one as excellent. All studies reported mean weight loss following energy deficit: the most severe was 4.0 kg. Protein dose during energy deficit varied from 0.5 g/kg/day to 2.4 g/kg/day. Six studies reported mean reductions in fat mass, with the largest being 4.5 kg. Four studies reported mean reductions in fat-free mass, while two studies reported an increase. Results support the recommendation that greater than 0.8 g/kg/day is necessary to mitigate the impact of energy deficit on a decline in lean body mass, while intakes up to 1.6 g/kg/day may be preferred. However, exact recommendations cannot be inferred as the severity and duration of energy deficit varied across studies. Longer and larger investigations are needed to elucidate protein’s role during energy deficit in active-duty military.

## 1. Introduction

Active-duty military personnel make up a unique and ever-growing population in all regions of the world. Similar to athletes, these individuals are often subjected to intense physical stress [1,2]. This may be in the form of structured training (such as basic training) or when engaged in combat or other arduous tasks during deployment. Sports nutrition is a growing field, with research centered on improving athletic performance. As such, athletes, especially professional athletes, are often afforded top-notch nutrition (both supplements and whole foods) to limit unhealthy energy deficits that may arise from intense training and competition in order to promote recovery and fat-free mass (FFM) optimization [3]. Military personnel, on the other hand, are often subjected to inadequate energy intake during periods of intense physical activity, as proper nutrition may be difficult to attain [4]. This is especially true in combat situations that often take place in rural areas where access to food can be scarce, where there may not be enough time to prepare meals, or where carrying food may be impacted by space and weight limitations that often place a greater importance on combat supplies and equipment [2,5]. This produces an energy deficit (ED) that can result in changes in body weight and composition that may hinder performance and recovery if not adequately managed [6].

Attaining an ideal body weight is important for active-duty military to best perform in tactical situations. Unfortunately, recent years have seen a greater prevalence of obesity in the military, adversely impacting military readiness and increasing the financial burden on the Department of Defense [7]. Excessive body weight is also hurting recruitment efforts, with 27% of American adults between 17 and 24 years old unable to serve in the military due to excess weight, which is now the leading medical reason individuals are withheld entry into the US military [7,8]. This has prompted many weight-loss programs and resources for military personnel designed to promote adherence to an ED induced by training and dietary restriction [8,9]. Although such an ED is needed to reduce fat mass (FM), prolonged or excessive EDs can decrease FFM, incite perturbations in many physiological responses, and impair performance [10,11]. In general weight loss trials, individuals classified as overweight-to-obese display decreases of FM and FFM (75% fat, 25% fat-free) [12]; however, FFM losses can be far greater when normal-weight individuals embark on high levels of physical activity that induce a sustained ED [13]. The importance of adequate dietary protein intake has been implicated in FFM optimization and performance when engaging in high levels of exercise [13,14,15]. Protein would therefore likely be an important dietary component of active-duty military personnel.

The most recent updates to the United States Military’s *Nutrition and Menu Standards for Human Performance Optimization* provide Military Daily Reference Intakes (MDRI) for energy intake (EI) and macronutrients [16]. It is recognized that EI will vary largely due to physical activity demands and climate, with the MDRI stating that EI for the average-sized military man (85 kg) or woman (69 kg) who is moderately active is 3400 or 2300 kcal/day, respectively, which can be increased by up to 125% during heavy activity. The MDRI for protein intake during long periods of physical activity or energy imbalance is 0.8 to 1.6 g/kg of body weight and should make up 10–35% of daily EI [16]. The present review analyzes the current research investigating the efficacy of increasing dietary protein intake during an ED, specifically focusing on body weight regulation (FM loss and FFM gain) among active-duty military personnel while also evaluating if enough evidence exists to provide specific recommendations for protein intake, identifying gaps in the literature where further research is needed.

## 2. Materials and Methods

We conducted a systematic review in accordance with the Preferred Reporting Items for Systematic Reviews and Meta-Analyses Checklist and included all relevant PRISMA checklist items [17]. The review methodology was defined prior to the conduct of the review and registered on PROSPERO (registration number CRD42023452177).

### 2.1. Search Strategy

A search for studies published from inception to 3 June 2023 was conducted using the PubMed and Web of Science database. The identification of studies used combinations of the search terms and were searched for in the Mesh database. Search terms included (military OR deployment OR combat OR soldier OR army OR ‘field training’ OR ‘field exercise’) AND (dietary protein intake OR dietary protein OR protein supplementation) AND (‘body composition’ OR ‘body mass’ OR ‘body weight’ OR ‘weight loss’ OR ‘weight change’ OR ‘weight reduction’ OR ‘muscle mass’ OR ‘fat free’ OR ‘fat mass’ OR ‘body fat’ OR ‘fat percentage’ OR ‘energy expenditure’ OR ‘energy balance’). The search was limited to human studies published in the English language in peer-reviewed journals and then filtered to only include randomized controlled trials, clinical trials, evaluation studies, and pragmatic clinical trials. The title, abstract, and keyword fields were assessed for general relevance and then the full text of remaining articles was analyzed for compliance with eligibility criteria. Furthermore, a hand search and supplemental forward citation search strategy was implemented to identify potential additional studies.

### 2.2. Study Selection and Data Extraction Eligibility Criteria

Relevant studies were included in this review if they (1) used a sample of active-duty military, (2) investigated the role of protein on body weight regulation during an energy deficit (ED), and (3) included an outcome measure for body weight (BW) and protein intake.

One reviewer independently extracted data specific to the following: population (population, recruitment, and type of military setting); intervention (study design, description, and dose); comparator (control, no intervention, or delayed intervention); outcome measures (body weight, physical activity, protein, and diet); and results (main findings, interpretations, conclusions, and funding).

### 2.3. Quality and Risk of Bias Assessment

Quality assessment of included studies was conducted utilizing the Downs and Black Checklist, a tool for assessing the risk of bias for randomized and non-randomized trials [18]. The checklist comprises 27 questions on reporting, external validity, internal validity, and power, with a maximum possible score of 28 for randomized studies and 25 for non-randomized studies. In accordance with others, the Downs and Black power question was modified for this review, 0 = not reported and 1 = reported, rather than the 5-point scale [19]. Downs and Black scores were assessed via the following range cut-points: excellent (26–28), good (20–25), fair (15–19), and poor (≤14) [20]. Missing data for extraction were marked as ‘unable to determine’ on the Downs and Black Checklist.

### 2.4. Data Analysis

Due to the heterogeneity of study measures and the limited number of included articles, results were not comparable; therefore, we were unable to perform a meta-analysis. Therefore, results are described narratively and focus on study population characteristics, design, quality, and the reported effect on body weight regulation in active-duty military during an energy deficit.

## 3. Results

### 3.1. Search Results

Our literature search yielded 284 articles, after exclusion of 261 studies due to title, abstract, or study design; 23 full text articles were evaluated for eligibility in accordance with the study’s selection criteria (Figure 1). Following evaluation, 17 full text articles were then excluded due to not meeting one or more of the criteria. Additionally, two studies [2,21] were identified from forward citation searches of two reviews [4,22] and deemed eligible for inclusion. In all, eight studies met each inclusion criterion and were retained for analysis [1,2,10,13,21,23,24,25].

### 3.2. Study Characteristics

In all eight included studies (Table 1), all participants were male [1,2,10,13,21,23,24,25]. Included studies varied in military training status, location, tasks, and branch (see Appendix A for a complete list of participant characteristics: military branch, duty duration, and additional key characteristics). The dates of publications ranged from 1987 to 2018. The duration of ED varied between studies from 8 days [24] to 6 months [2]. Moreover, the measurement for confirming ED was diverse, with five studies utilizing doubly labeled water (DLW) [1,2,10,23,24], two studies using total energy expenditure estimations based on previous work [21,25], and one study using indirect calorimetry and controlled feeding [13]. Protein source was directly provided to participants in all the included studies except for one [2]. In two studies, meals ready to eat (MREs) were exclusively used for protein feeding [10,21], three studies were supplemented with protein [23,24,25], two provided meals via pre-planned menus [1,13], and ad libitum protein intake was measured in one study [2].

### 3.3. Quality Assessment

The quality of included articles can be found in Table 2 and the full checklist with scoring is available in Appendix A. Five studies were classified as good, two as fair, and one as excellent [1,2,10,13,21,23,24,25]. The excellent study by Berryman et al. only missed two checklist requirements due to not reporting if recruited subjects were representative of the source population [23]. The average reporting score for articles was excellent (9.7/11 points), while internal validity–bias (5.6/7 points) and internal validity–confounding (4.5/6 points) were good. However, external validity (1.1/3 points) was scored only fair in quality. All studies included or stated that a power calculation was conducted prior to implementation [1,2,10,13,21,23,24,25]. Importantly, all studies met the requirements for using appropriate statistical tests to assess main outcome measures. Additionally, investigator choices for validity, accuracy measures, and outcome measures were also scored as appropriate [1,2,10,13,21,23,24,25]. Three of the studies were non-randomized [2,10,21], while only one study provided a valid description of the source population [2]. Seven of the eight included studies reported the funding source for the project, five of which were supported by the U.S. Army Medical Research [1,10,13,21,23] and one by the United Kingdom Ministry of Defense [2]. One study reported that it was ‘funded by a technologies in metabolic monitoring grant’ [24]. The one study that did not report funding acknowledged support from several supplement manufacturers [25].

### 3.4. Primary Outcomes

Table 3 provides a summary of results.

#### 3.4.1. Protein and Weight Loss

All included studies reported some degree of weight loss among participants [1,2,10,13,21,23,24,25]. Alemany et al. and Askew et al. both reported the greatest decrease in mean body weight at 4.0 kg [21,24], while McAdam et al. had the least at 0.2 kg [25]. Regarding protein dose during ED, the largest was investigated by Pasiakos et al. at 2.4 g/kg/day [13] and the smallest at 0.5 g/kg/day by Alemany et al. [24]. Three studies provided additional protein to participants in supplement form [23,24,25]. Of these, Berryman et al. [23] provided the largest dose at 3.5 g/kg/day during a refeed period following an 18-day ED. In another, McAdam et al. [25] provided a 38.6 g whey supplement two times a day to one group of soldiers during Initial Entry Training (IET), while Alemany et al. [24] provided an 18 g protein drink and 11 g protein bar in addition to the groups’ pre-planned standardized MRE diet.

#### 3.4.2. Protein and Changes in Fat Mass

Six included articles reported reductions in fat mass (FM) at final analysis [2,13,21,23,24,25]. Three of these studies used dual X-ray absorptiometry (DEXA/DXA) for body composition measurement [13,23,24], while two utilized 7-site-or-greater skinfold [2,25] and one implemented underwater weighing [21]. The largest reduction in mean FM, 4.5 kg, was observed by McAdam et al. in the whey protein (WP) group compared to 2.7 kg in the carbohydrate (CHO) group [25]. In the trial by Pasiakos et al., a difference in FM loss between groups was also observed as participants randomized to two (1.6 g/kg/day) and three times (2.4 g/kg/day) the recommended dietary allowance (RDA); both lost 1.9 kg compared to 1.6 kg in the RDA (0.8 g/kg/day) group [13]. Despite being in a 2318 kcal per day ED during an 8-day military field exercise, the smallest observed FM reduction was detected by Alemany et al. [24], with participants in the 0.9 g/kg/day group losing nearly the same amount of FM as those consuming 0.5 g/kg/day with FM reductions of 1.4 kg and 1.5 kg, respectively.

The most severe ED was examined by Berryman et al. [23], where participants in all three groups were, on average, in a deficit of 4203 kcal per day for seven days. This severity was achieved during US Marine Survival, Evasion, Resistance, Escape (SERE) training, where five days consisted of an intentional limited availability of food (300 kcal/d). Following the period of ED, participants were randomized to one of three groups for a 28-day REFED period, where total protein intakes were either CON (2.0 g/kg/day), MOD (3.2 g/kg/day), or HIGH (3.5 g/kg/day). Across all groups, mean FM decreased by 2.7 kg during ED, while at POST-REFED, FM reductions had increased by 2.3 kg, which resulted in no significant differences between PRE-SERE and POST-REFED measures of FM.

#### 3.4.3. Protein and Changes in Fat-Free Mass

Fat-free mass (FFM) was reported in six of the included articles (Table 1) [2,13,21,23,24,25]. Body composition was measured via dual X-ray absorptiometry (DEXA/DXA) in three studies [13,23,24], as 7-site-or-greater skinfold for two studies [2,25] and by underwater weighing in one study [21]. Increases in FFM were observed in two studies [21,25]. McAdam et al. [25] demonstrated that during an estimated ED of 595 kcal per day, the WP and CHO groups increased FFM by 4.2 kg and 3.6 kg, respectively, with a large effect size for both the WP (Cohen’s d = 0.44) and CHO (Cohen’s d = 0.42) groups. In the other, Askew et al. [21] reported a mean FFM increase of 0.2 kg, which was observed only in the group receiving 112 g protein per day, while the group receiving 64 g/day saw a mean FFM decrease of 1.5 kg. Importantly, the 112 g/day group was in a calculated ED of 467 kcal/d, while the 64 g/day group was in a 1946 kcal/d ED.

Decreases in FFM were observed in four studies [2,13,23,24]. In the previously discussed Berryman et al. [23], mean FFM was reduced by 3.1 kg at the end of SERE training. However, this decrement was resolved and returned to pre-SERE values after the end of the 27-day REFED period. Furthermore, this resolution of FFM was observed in all participants regardless of protein-dose allocation (CON, MOD, HIGH) during REFED. Pasiakos et al. [13] reported that participants randomized to receive either the 2x-RDA (1.6 g/kg/day) or 3x-RDA (2.4 g/kg/day) protein dose maintained a greater percentage of FFM compared to those in the RDA group (0.8 g/kg/day). Interestingly, no difference was observed between the 2x- and 3x-RDA protein groups, indicating a ceiling effect for protein intake. Alemany et al. [24] found that the protein-dose intervention of either 0.5 g/kg/day or 0.9 g/kg/day appeared to have no measurable effect on FFM maintenance over the duration of the study (*p* = 0.7), in as much as both groups lost a similar amount of mean FFM at 1.7 kg and 1.3 kg, respectively (*p* = 0.001).

At 6 months in length and with a study population of 750, the study by Fallowfield et al. [2] was the longest in duration and the largest sample to investigate FFM changes. Over the course of a 6-month deployment to Afghanistan, the authors observed a mean ED of 1043 kcal/d among Royal Marines as measured by DLW. Throughout the first half of deployment, mean FFM decreased by 1.9 kg but increased by 2.1 kg during the second half. Despite these FFM fluctuations, the percentages of protein intake at mid-deployment were consistent with pre- and post-intake percentages across the sample. In a subgroup analysis (*n* = 75), the authors discovered that mean FFM increased by 1.0 kg during the 14-day rest and recuperation (R&R) leave period which occurred mid-tour back in the United Kingdom (U.K.). During R&R, mean protein intake was calculated to be 125 g/day, which was statistically greater than the protein intake reported by those who did not return to the U.K. for R&R (*p* < 0.001). However, greater protein intake did not result in statistically different changes in FFM between subgroups.

### 3.5. Secondary Outcomes

#### 3.5.1. Protein Supplementation and Weight Maintenance

As discussed, three studies provided additional protein to participants in supplement form [23,24,25] via protein drinks and/or bars. Berryman et al. [23] investigated protein’s influence on body weight by randomly assigning participants to one of three groups (CON 6 g/day, MOD 66 g/day, HIGH 94 g/day). Alemany et al. [24] (0.5 g/kg/day, 0.9 g/kg/day) and McAdam (WP 38.6 g × 2/d, CHO 0.5 g × 2/d) et al. [25] implemented a two-group design, the latter of which was not randomized. No differences in FM or FFM were observed between groups in any of the three studies with regards to higher protein supplementation compared to carbohydrate [23,24,25].

#### 3.5.2. Protein Supplementation, Body Composition, and Optimization of Performance

The only protein-supplementation-included study to observe a difference in body composition was McAdam et al. [25], who reported a significant difference in FM at post-intervention after controlling for initial FM (*p* = 0.04). No statistically significant difference in FFM or body fat percentage between groups were observed in any included study. Regarding optimization of performance, McAdam et al. [25] found that the WP group significantly improved push-up performance compared to the CHO supplementation group (*p* = 0.002) but did not discover a difference between groups in the run or sit-up test.

As for the effect of protein supplementation on physiological measures to benefit performance, Alemany et al. [24] reported that both 0.5 and 0.9 g/kg/day protein dose attenuated insulin-like growth factor-1 (IGF-1) and decreased sex hormone-binding globulin (SHBG) but did not exert an effect on measures of anabolic growth factors. Berryman et al. [23] reported that supplementation contributed to 22% of total energy intake during the participant REFED period and that regardless of protein assignment, the majority of physiological insults (protein synthesis, protein breakdown, and amino acid perturbance) that occurred during severe ED had resolved at 27 days post, with the only notable exception being net protein balance.

## 4. Discussion

This systematic review investigated the effects of protein on body weight regulation in active-duty military during periods of ED. A total of eight studies met our inclusion criteria [1,2,10,13,21,23,24,25]. While the importance of protein during and around periods of ED has been investigated broadly and reviewed by others [14,15], this review identifies and describes the current state of the literature on this topic in the active-duty military population. Moreover, while the U.S. military and others have provided guidance for protein recommendations in active-duty personnel and reviewed the impact of combat rations on body weight regulation [4,11,22,26,27,28,29], few have reviewed protein’s role in body weight regulation during heavy combat and training.

In agreement with Carbone et al., Pasiakos et al., and others [30,31,32], the current review points to protein intakes exceeding the RDA (0.8 g/kg/day) to be necessary to minimize FFM loss during ED. Additionally, McAdam et al. found that additional protein via supplementation may be beneficial for improving body composition and performance in military personnel [25]. These findings are in agreement with others who have identified the benefits of protein supplementation on athletic performance [33,34]. However, three studies included in this review showed that additional protein provided to participants via supplementation did not improve body weight maintenance more than an energy-matched carbohydrate supplement [23,24,25].

Since military personnel are subjected and expected to perform optimally during periods of ED relying heavily on MREs and supplements, it is vital that adequate protein is both palatable and available during training and combat [11,22,26,35]. Additionally, as Berryman et al. demonstrated [23], even during periods of extreme energy deficit (300 kcal/d), if adequate protein and total energy intake is ingested for 4 weeks during recovery, FFM is likely to reach pre-ED values.

The major strength of this review is the gathering of research investigating protein’s role in body weight regulation among active-duty military during periods of energy deficit. Instead of reviewing military guidelines and recommendations, this review focused on the effect of protein dose on body weight, FFM, FM, and performance. Included studies provided positive evidence that daily protein consumption greater than the RDA of 0.8 g/kg is necessary to mitigate FFM loss during ED and that 1.6 g/kg would likely be ideal. However, reported body composition results should be interpreted with caution as different methods for measurement were used across studies. Additionally, it provides worthwhile information on the protein doses needed for improving active-duty military performance during and after heavy training and/or combat. The primary limitation of this review is that the included studies varied in their length and severity of ED; thus, we were unable to run a meta-analysis to pinpoint an exact protein intake recommendation. Despite our best efforts to provide more rigorous analysis across all included articles, the variations in measure duration and study design render this unattainable. Likewise, caution should be taken when considering our risk of bias assessment as our scoring reflects our reviewers’ subjective views. While all but one study was scored as fair [21], three studies were non-randomized [2,10,21] and only one study included a description of the source population [2]. However, all included articles implemented the appropriate statistical tests to assess the main outcome measures while also being sure to conduct a power analysis [1,2,10,13,21,23,24,25].

An additional limitation of this review is the lack of information included regarding micronutrient intake. Since this information was limited, it can be assumed that it was likely below RDA values during periods of ED and thus could be a confounding variable in the effects of an ED on body composition and associated protein recommendations.

As noted, the present review provides evidence for a protein intake above the RDA being beneficial for military personnel during times of ED. However, study duration and sample size for the limited number of studies included in this review limit our ability to assess dietary protein requirements. Larger, longer, and more precise investigations should be explored in the future to better understand protein’s role in body weight regulation and FFM maintenance during and after heavy training, combat, and ED. Future research standardizing ED across individuals and manipulating protein intake is needed to provide a precise protein intake recommendation. Additionally, while Wardle et al. [36] began the conversation on considerations for sex-specific nutrition among military personnel, the lack of females included in any of the included articles highlights a gap that ought to be considered in future trials to better understand any deviations in protein-dose sex effects.

## 5. Conclusions

This systematic review assembled the current body of evidence investigating the role of protein on body weight regulation, FM, FFM, and body composition in active-duty military during ED. Additionally, it sought to evaluate if enough evidence exists to provide specific recommendations for intake in active-duty military, identify gaps in the literature, and expose where future research is most needed. Evidence suggests that protein consumption greater than the RDA (0.8 g/kg/day) is necessary to mitigate detrimental reductions in FFM during ED of 3 days or greater and that 1.6 g/kg/day is likely necessary to offset the detrimental impact of ED on FFM. Additional protein intake via supplementation may also be beneficial for improvements in body composition and military performance measures. However, the number of articles meeting the inclusion criteria was limited, and additional research is needed to better elucidate the optimal daily protein dose for active-duty military during combat and training. To ensure adequate protein is consumed, it is advised that future active-duty military investigations explore methods to increase the palatability and availability of higher protein-dose MREs, supplements, and foods during ED.

## Figures and Tables

**Figure 1 nutrients-15-03948-f001:**
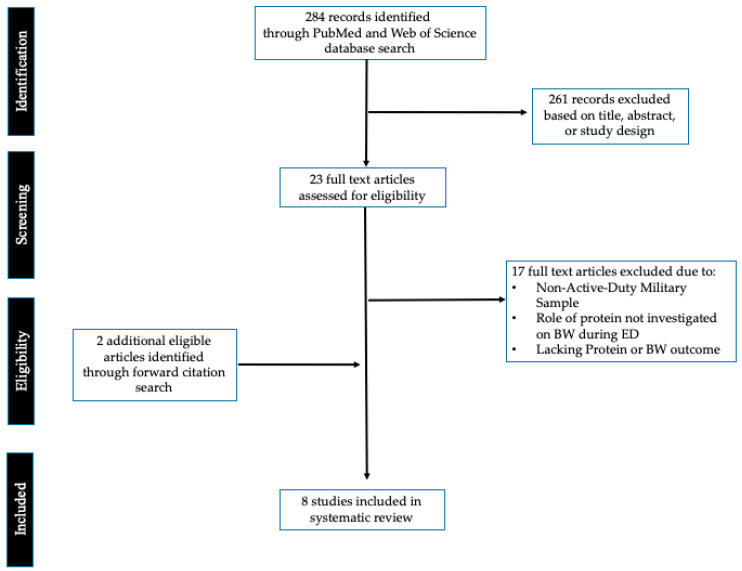
Preferred Reporting Items for Systematic Reviews and Meta-Analyses (PRISMA) flow diagram of study selection.

**Table 1 nutrients-15-03948-t001:** Characteristics of included articles (*n* = 8).

Reference, Year	Country	Type of Study	Final Sample (*n*)	Participants	Time Period	Study Length	Age (Yrs.)	Protein Dose	EI Measure	Body Weight Measure
Askew et al., 1987 [21]	U.S.A.	Experimental between groups (Pre/Post)	34	2nd and 3rd Battalions	September 1986–October 1986	30 days	27	112 g/day vs. 64 g/day	Self-report and nutrient analyses system (USARIEM), calculated from TEEE	BW (SECA Scale), FFM and FM (underwater weighing)
Alemany et al., 2008 [24]	U.S.A.	Mixed-model, repeated measures	34	U.S. Marine Corps infantry officer candidates	2006	8 days	24.5 ± 0.3	0.9 g/per kg/d vs. 0.5 g/per kg/d	Observed (wrappers) and MRE nutrient database, DLW	BW (digital scale), FFM, FM (DXA)
Berryman et al., 2016 [23]	U.S.A.	RCT	63	U.S. Marines	January 2014–March 2015	45 days	25 ± 3	7 g/day, 84 g/day, or 133 g/day supplement	24 h dietary recall and database analysis, DLW	BW (Digital Scale), FFM, FM (DEXA)
Booth et al., 2003 [1]	Australia	Experimental btw groups	37	Airfield defense guards	April 1999	12 days	22	63 g/day, 88 g/day, or 116 g/day	Observed (wrappers) and database analysis, DLW	BW (SECA Scale), FFM and FM (BIA)
Fallowfield et al., 2004 [2]	U.K.	Within-subject, repeated measures	176	Royal Marines stationed in U.K. for deployment to Afghanistan	March 2010–October 2010	6 months	28 ± 7	125 g/day vs. 95 g/day	Self-report (food diary) and analysis, DLW	BW (Scale), Body Comp (skinfolds (eight sites) and circumferential girths (six sites))
Margolis et al., 2014 [10]	Norway	Longitudinal observational	21	Norwegian conscripted soldiers	2012	7 days	20 ± 1	1.59 g/kg/day, 1.71 g/kg/day	Observed (wrappers and food logs) and analysis, DLW	BW (Digital Scale)
McAdam et al., 2018 [25]	U.S.A.	Repeated measures, double-blind, parallel groups	69	U.S. Army soldiers	2017	8 weeks	19 ± 1	0.5 g vs. 38 g (2xd protein supplement)	Self-report (diet recall), ED estimation from previous work	BW (Scale), FFM, FM (7-site skinfold)
Pasiakos et al., 2013 [13]	U.S.A.	RCT	39	Military personnel from the U.S. Army	2012	31 days	21 ± 1	0.8 g/kg/day, 1.6 g/kg/day, or 2.4 g/kg/day	Objective (dietitian/metabolic kitchen prepared meals), indirect calorimetry	BW (digital scale), FFM, FM (DXA)

BW, body weight; FM, fat mass; FFM, fat-free mass; Body Comp, body composition; RCT, randomized controlled trial; UK, United Kingdom; U.S., United States of America; BIA, bioelectrical impedance analysis; DEXA/DXA, dual X-ray absorptiometry; ED, energy deficit; DLW, doubly labeled water.

**Table 2 nutrients-15-03948-t002:** Quality of included articles.

First Author, Year Maximum Points Available	Reporting 11 Points	External Validity 3 Points	Internal Validity–Bias 7 Points	Internal Validity–Confounding 6 Points	Power 1 Point	Total	Quality Rating α
Randomized studies (maximum score = 28)							
Alemany et al., 2008 [24]	11	1	6	5	1	24	Good
Berryman et al., 2016 [23]	11	1	7	6	1	26	Excellent
Booth et al., 2003 [1]	10	1	5	5	1	22	Good
McAdam et al., 2018 [25]	9	1	7	6	1	24	Good
Pasiakos et al., 2013 [13]	11	0	5	5	1	22	Good
Non-randomized studies (maximum score = 25)							
Askew et al., 1987 [21]	9	1	5	3	1	19	Fair
Fallowfield et al., 2004 [2]	9	3	5	3	1	21	Good
Margolis et al., 2014 [10]	8	1	5	3	1	18	Fair

Note: Assessed by the Downs and Black Checklist [18]. α Quality rating: Excellent (26–28), Good (20–25), Fair (15–19), and Poor (≤14).

**Table 3 nutrients-15-03948-t003:** Summary of results from included articles (*n* = 8).

Reference, Year	Protein Groups	ED Length	TDEI (kcal)	TDEE (kcal)	ED (kcal/d)		BW (kg)		FM (kg)		FFM (kg)		BF%		Key Findings
			Mean	Mean	Mean		Mean	SD	Mean	SD	Mean	SD	Mean	SD	
Askew et al., 1987 [21]	MRE VI 112 g/day	30 days	2782 of 3600	3250	−467	Pre Post %Change	74.8 73.6 −1.6	2.1 1.7	11.9 10.4 −12.6	1.0 0.9	62.9 63.1 0.3	1.0 1.9	15.6 13.9 −10.9	0.8 1.0	Decrease in FFM for RLW-30 *
	RLW-30 64 g/day		1946 of 1976	3275	−1946	Pre Post %Change	79.3 75.3 −5.0	1.8 1.5	13.2 10.6 −19.7	1.0 0.8	65.8 64.3 −2.3	0.9 0.8	16.5 13.9 −15.7	0.8 0.7	
Alemany et al., 2008 [24]	MRE 0.9 g/kg/day	8 days	1530 *ρ*	3800/d *ρ*	−2318 *ρ*	Pre Post %Change	83.0 79.0 −4.8 α	9.6 8.9	13.9 12.5 −10.1 α	3.1 3.8	69.2 67.5 −2.4 α	4.7 8.3	16.6 15.3 −7.8α	3.6 4.2	Protein dose (O) effect on FFM loss
	MRE 0.5 g/kg/day		1530 *ρ*	3944/d *ρ*	−2318 *ρ*	Pre Post %Change	81.6 78.5 −3.8 α	5.9 5.7	12.5 11.0 −11.0 α	3.2 2.7	69.1 67.8 −1.8 α	8.3 7.7 −1.9 α	15.1 13.3 −11.9 α	3.3 3.0	
Berryman et al., 2016 [23]	CON 6 g/day Sup, 2.0 g/kg/day Total	7 days	300, REFED (4506)	NR	−4203	Pre Post %Change	85.3 85.2 −0.1 α	8.4 7.5	NR 14.2	NR 3.5	NR 67.4	NR 5.4	NR	NR	CON, MOD, or HIGH Sup (O) on any outcome measure POST-REFED
	MOD 66 g/day Sup, 2.0 g/kg/day Total		300, REFED (4612)	NR	−4203	Pre Post %Change	83.1 83.3 +0.2 α	11.0 10.1	NR 13.7	NR 4.5	NR 66.2	NR 7.7	NR	NR	
	HIGH 94 g/day Sup, 3.5 g/kg/day Total		300, REFED (4337)	NR	−4203	Pre Post %Change	83.0 83.1 +0.1 α	8.8 8.0	NR 12.8	NR 3.2	NR 66.9	NR 6.4	NR	NR	
Booth et al., 2003 [1]	Full CRP 88 g/day		2200 of 3897	3650ε	NR	Pre Post %Change	76.1 74.6 −1.7	10 8.6	NR NR	NR NR	NR NR	NR NR	16 12.7 −20	7.2 6.1	1800 kcal/d ration, 54% Carb, 16% Protein, 30% Fat was sufficient to maintain nutritional status (O)
	One-half CRP 63 g/day		1600 of 2155	3650 ε	NR	Pre Post %Change	72 70.1 −2.6	8.8 8.1	NR NR	NR NR	NR NR	NR NR	14.3 11.4 −18	4.0 4.0	
	Fresh 116 g/day		2850 of 3600	3650 ε	NR	Pre Post %Change	83 82	10.5 10.5	NR NR	NR NR	NR NR	NR NR	19.5 18.2 −2.0	5.2 6.3	
Fallowfield et al., 2004 [2]	Repeated measures (Pre, Mid, Post)	6 months	Pre 3033 κ Mid 2531 κ Post 2685 κ	3626 κ	−1043 κ	Pre Mid Post %Change	82.4 78.5 80.9 −1.8 α	9.1 8.0 8.3	NR NR NR −1.7 β	NR NR NR	NR NR NR −1.9 β	NR NR NR	17.2 15.9 16.0 −7.0 α	4.9 4.6 4.2	BW, FM, FFM and BC decrease *, FFM increase Post
Margolis et al., 2014 [10]	MTT 1.59 g/kg/day	4 days	3098 of 3800	5480	−2382	Pre Post %Change	82.7 80.5 −2.6 α	9.7 8.1	NR NR	NR NR	NR NR	NR NR	NR NR	NR NR	2× RDA protein not sufficient for protein balance maintenance during severe ED
	SKI 1.71 g/kg/day	3 days	3461 of 5100	6851	−3390	Pre Post %Change	80.5 80.2 −0.4 α	8.1 7.9	NR NR	NR NR	NR NR	NR NR	NR NR	NR NR	
McAdam et al., 2018 [25]	WP 38.6 g x2/d Sup, 2.8 g/kg/day Total	8 weeks	Pre-NS 2825 Post-NS 2930 Post-SI 3516	NM	−595 Est.	Pre-NS Post-SI %Change	73.4 73.2 −0.2 α	12.7 10.5	13.5 8.9 −34.0 α	6.1 4.2	60.0 64.2 +7.0 α	7.9 7.5	NR NR	NR NR	WP—Push up (+), FM (−) * compared to CHO. FFM, sit-up, run (O)
	CHO 0.5 g x2/d Sup, 1.6 g/kg/day Total		Pre-NS 2624 Post-NS 2766 Post-SI 3348	NM	−595 Est.	Pre-NS Post-SI %Change	72.3 73.2 +1.2 α	10.9 7.9	12.2 9.5 −22.1 α	6.1 3.9	60.1 63.7 +6.0 α	7.3 6.1	NR NR	NR NR	
Pasiakos et al., 2013 [13]	RDA 0.8 g/kg/day	21 days	1883	485 ε	−40% ψ	Pre Post %Change	78 74.5 −4.5 α	3 ^ 0.3 ^	NR NR −1.6 β	NR NR	NR NR −2.3 β	NR NR	NR NR −1.3 β	NR NR	2× RDA sparred FFM, decrease FM compared to RDA
	2× RDA 1.6 g/kg/day		1820	498 ε	−40%ψ	Pre Post %Change	76 73.3 −3.6 α	3 ^ 0.2 ^	NR NR −1.9 β	NR NR	NR NR −0.8 β	NR NR	NR NR −1.8 β	NR NR	
	3× RDA 2.4 g/kg/day		1766	498 ε	−40% ψ	Pre Post %Change	77 73.7 −4.3 α	2 ^ 0.3 ^	NR NR −1.9 β	NR NR	NR NR −1.2 β	NR NR	NR NR −1.9 β	NR NR	

ED, energy deficit; TDEI, total daily energy intake; TDEE, total daily energy expenditure; BW, body weight; FM, fat mass; FFM, fat-free mass; BF%, percentage body fat; BC, body composition; NM, not measured; RLW, ration, lightweight; MRE, meal, ready-to-eat, Sup, supplement; Est, estimated; Pre-NS, pre-nutrition supplement; Post-NS, post-nutrition supplement; Post-SI, post-supplement intervention; α = calculated as (*V*2 − *V*1)|*V*1| × 100; * = statistical significance; (O) = no effect/difference observed between groups; (+) = increase; (−) = decrease; *ρ* = reported as megajoules; κ = reported as kilojoules; ε = reported as daily exercise energy expenditure; β = reported as mean difference; ^ = reported as SEM; ψ = reported as % of total energy balance.

## Data Availability

Since this was a review, no new data was generated. All data used is included in manuscript.

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
