# Peer review of "The Role of Dietary Protein in Body Weight Regulation among Active-Duty Military Personnel during Energy Deficit: A Systematic Review"

_nutrients, 2023, doi:10.3390/nu15183948_

Round 1
Reviewer 1 Report
The study presented reviewed actual work (observational and intervention studies) focusing on energy and protein intake of military personnel on duty. The literature search was performed following internationally accepted principles. There are, however, some pitfalls and misleading statements which should be corrected/clarified.
1. The idea of the authors was to „analyze the current research investigating the efficacy of increasing dietary protein intake during an ED period..“. Unfortunately, none of the studies selected followed a clear and controlled study design to evaluate different protein intakes. Most of the studies simply reported actual energy and protein intake. In addition, there is no information given how this protein effect can be measured in these populations. Since the ED varied in a broad range and the period of ED was from days to months, no reliable conclusions can be drawn. Indeed, the authors themselves mentioned that „.. due to the heterogeneity of study measures …. results were not comparable. So, what is the scientific benefit of this compilation? And why this article can be qualified as „critical“ review?
2. Unfortunately, the authors did not report how the body composition (FM, FFM etc.) has been measured in the studies presented. In addition, important information on nutroent intake (minerals, vitamins etc.) are not given. It ist o assume that the intake of micronutrients is also limited and probably below reference values. Then, any comments on the effects of nutrient intervention on body composition etc. is not allowed.
3. Ref. 17: Obviously, the goal of defining these standard values ist …human performance optimization“, not body composition, e.g., lean body mass. Which criteria have been used to evaluate performance? And how these reference values are related to the goals of the present review?
4. As mentioned above, reliable conclusions on an „optimized“ or „desired“ protein intake cannot be drawn.
Author Response
comments address in attached file

Reviewer 2 Report
A review and titled: The role of dietary protein in body weight regulation among active-duty military personnel during energy deficit: A Critical Review.
The objective was to analyze the current research investigating the efficacy of increasing dietary protein intake 77 during an ED, specifically focusing on body weight regulation (FM loss and FFM gain) 78 among active-duty military personnel. Review investigated a novel, original and relevant topic. Comments and suggestions to strengthen the manuscript are presented below
Keywords: I suggest that all keywords be searched in the Mesh database (https://www.ncbi.nlm.nih.gov/mesh/)
Abstract: The abstract does not have justification and objective of the study
Introduction: The introduction was clear. I suggest finding more arguments to justify this study. Possible benefits of doing a meta-analysis and meta-analysis can be indicated.
Methods: On the other hand, review is not registered in a database of systematic reviews and meta-analyses (for example: PROSPERO https://www.crd.york.ac.uk/prospero/ ). It must be explained why it is not registered.
It is not understood because only PubMed database was searched and not in other databases, for example: Central by Cochrane, Scopus, Web of Science.
Discussion: Authors discussed the results and how they can be interpreted in perspective of previous studies.
Although, little is said about the findings and their implications. I suggest delving into the implications and practical contributions of the results.
In addition, Future research directions may also be mentioned.
Author Response
Comments addressed in attached file

Round 2
Reviewer 1 Report
Thanks for the revisions made. The goals are now adequately presented and the obvious limitations arediscussed.y
Reviewer 2 Report
The authors took all comments into account. Now the new corrected version of the manuscript reads more clearly and accurately. I agree that this new version is published